# A New Perspective on Thyroid Hormones: Crosstalk with Reproductive Hormones in Females

**DOI:** 10.3390/ijms23052708

**Published:** 2022-02-28

**Authors:** Bingtao Ren, Yan Zhu

**Affiliations:** 1School of Pharmacy, Fudan University, Shanghai 200032, China; btren19@fudan.edu.cn; 2Laboratory of Reproductive Pharmacology, NHC Key Laboratory of Reproduction Regulation, Shanghai Institute for Biomedical and Pharmaceutical Technologies, Fudan University, Shanghai 200032, China

**Keywords:** thyroid hormones, genomic effect, nongenomic effect, reproductive hormones

## Abstract

Accumulating evidence has shown that thyroid hormones (THs) are vital for female reproductive system homeostasis. THs regulate the reproductive functions through thyroid hormone receptors (THRs)-mediated genomic- and integrin-receptor-associated nongenomic mechanisms, depending on TH ligand status and DNA level, as well as transcription and extra-nuclear signaling transduction activities. These processes involve the binding of THs to intracellular THRs and steroid hormone receptors or membrane receptors and the recruitment of hormone-response elements. In addition, THs and other reproductive hormones can activate common signaling pathways due to their structural similarity and shared DNA consensus sequences among thyroid, peptide, and protein hormones and their receptors, thus constituting a complex and reciprocal interaction network. Moreover, THs not only indirectly affect the synthesis, secretion, and action of reproductive hormones, but are also regulated by these hormones at the same time. This crosstalk may be one of the pivotal factors regulating female reproductive behavior and hormone-related diseases, including tumors. Elucidating the interaction mechanism among the aforementioned hormones will contribute to apprehending the etiology of female reproductive diseases, shedding new light on the treatment of gynecological disorders.

## 1. Introduction

The butterfly-shaped thyroid gland is an endocrine organ located below the thyroid cartilage in the mammalian neck [1,2]. The thyroid gland is responsible for secreting thyroid hormones (THs), including triiodothyronine (T3) and thyroxine (T4), both of which are mainly generated by thyroid follicular epithelial cells [3]. The synthesis and secretion of THs are regulated by thyroid-stimulating hormone (TSH) released by the pituitary gland [4]. T4 accounts for approximately 80% of the total THs, whereas T3 only accounts for about 20%. In circulation, T4 is converted to active T3 by type I (D1) and type II (D2) deiodinases located in the liver, kidneys, muscles, and thyroid glands [5,6]. Approximately 80% of T3 is produced by deiodination of T4 [7]. Meanwhile, THs are negatively feedback regulated by TSH and thyrotropin-releasing hormone (TRH) to maintain normal TH levels in the blood [8,9].

After entering into the blood, THs are rapidly transported to target tissues, where they facilitate the body’s utilization of energy and support the functions of vital organs, including maintaining the normal function of the reproductive system. Genomic and nongenomic mechanisms are involved in the modulation of THs. Here, we focused on elucidating the patterns of interaction among THs, reproductive hormones, and their receptors and assessing the genomic and nongenomic effects on these interactions, as well as the possible clinical significance of this crosstalk, providing new insights into the clinical diagnosis and treatment of reproductive diseases. 

## 2. Genomic Effect of THs

T3 enters cells through the assistance of monocarboxylate transporter 8/10 (MCT8/10) and then binds to thyroid hormone receptors (THRs) in the nucleus to exert its cellular effect, also called the genomic effect [10,11,12,13]. THRs belong to the steroid–thyroid hormone nuclear receptor superfamily [13,14]. The THRs proteins are encoded by the *THRA* and *THRB* genes [15]. Thyroid hormone receptor alpha (THRA) protein is cleaved into THRα1, THRα2, THRα3, THRΔα1, THRΔα2, and THRα-ΔE6 via alternative splicing [13,16]. Of them, THRα1 is a genuine nuclear receptor which directly induces an inhibitory effect in the absence of T3 or transcriptional activation after binding to T3. The other THRα variants are non-receptors lacking the T3-binding domain [16,17,18]. For example, THRΔα1 is regarded as a candidate mediator that binds to T4 in the cytoplasm to mediate nongenomic effects, rather than binding to T3 to mediate genomic effects [19]. In contrast, the predominant isoforms of thyroid hormone receptor beta (*THRB*) genes are cleaved into THRβ1, THRβ2, THRβ3, and THRβ4, all of which are nuclear receptors and bind to T3 with high affinity to regulate genomic mechanisms [16,20,21]. 

Genomic effects mediated by THRs are dependent on the status of TH ligand. In the absence of THs, THRs combine with the thyroid-hormone response element (TRE) in the form of a monomer or homodimer and recruit corepressors, e.g., nuclear receptor corepressor (NCoR) and silencing mediator of retinoic acid and thyroid hormone receptor (SMRT), to deacetylate histone proteins, producing a tighter conformation of chromatin and inhibiting transcription [13,22,23,24]. However, in the presence of TH, the ligand-bound THRs couple with TRE as a heterodimer with retinoid X receptor (RXR) and then recruit coactivators, e.g., steroid receptor coactivator (SRC-1), and CREB-binding protein (CBP), as well as acetylate histones, which creates a more open chromatin structure, promoting gene transcription [25] (Figure 1). 

TRE is a short DNA sequence that is located in gene regulatory regions and is indispensable for mediating the genomic effects of T3. TRE shares a consensus DNA sequence with hormone response elements (HREs) of members of the steroid hormone receptor superfamily, such as estrogen receptors (ERs), peroxisome proliferator-activated receptors (PPARs), retinoid X receptors (RXRs), and vitamin D receptors (VDRs) [26,27,28,29]. In Table 1, we list the members of the steroid and TH nuclear receptor superfamilies that share the same DNA consensus sequence and function in the female reproductive system. TRE is recognized by the members of the superfamilies and accumulates near the target gene promoter to activate or inhibit transcription [26,29]; the ability of TRE to respond to other steroid hormone receptors enables the crosstalk between THRs and steroid hormone receptors, providing multiple selectivity for gene regulation. The crosstalk is subsequently involved in the regulation of genes that share the same downstream biological process and pathway or overlapping target genes [30]. 

In addition, several studies have indicated that THR induces the crosstalk with other steroid hormone receptors through incomplete hormone response elements or coactivators, including androgen receptors (ARs), glucocorticoid receptors (GRs), mineralocorticoid receptors (MRs), and progesterone receptors (PRs). The genomic crosstalk of these receptors at the DNA and transcription levels is another interactive pattern between THs and steroid hormones. 

THs and steroid hormones are lipid-soluble hormones that directly bind to their receptors in the nucleus and act as transcription factors to mediate intracellular effects [45]. Notably, continuous intracellular crosstalk is induced by the joint action of THs and steroid hormones on target cells [46]. The receptors for THs and steroids are nuclear receptors that exert modulatory function mainly via genomic effects. 

### 2.1. Crosstalk between THs and Estrogen 

ERs belong to the steroid hormone receptor superfamily [47]. Estrogen response element (ERE), the DNA binding site of ERs, is highly similar to TRE, and both of them have the same DNA-binding sequence AGGTCA [48], as shown in Table 1. THα1 is able to competitively bind to ERE in the cell nucleus, inhibiting the recruitment of co-activator essential to ERs-medicated transcription, resulting in the suppression of estrogen-induced transcriptional activation [27,28,48,49]. ERs bind to TRE to mediate a strong estrogen-dependent transcriptional activation in the absence of THRs [26]. Nevertheless, in the presence of THRs, THR preferentially binds to TRE and blocks the entry and activation of other receptors, such as ERs. Under this condition, the binding of ERs with TRE is impeded, and they bind to their response elements to activate transcription [26]. This type of crosstalk considerably improves the fidelity of the ligand-specific response. Therefore, the target gene can be regulated by the ERs in the absence of THRs [26] via compensatory transcription, which is a process automatically initiated by the body for adaptation to the lack of THRs.

In another study, mutated THRs were found to lack the sites that bound to the AGGTCA sequence, but they were still able to inhibit the transcriptional response of estrogen [28]. Researchers assumed that the interaction between ERs and THRs was likely influenced by protein levels, although the existence of competition between ERs and THRs at the DNA level could not be excluded. Subsequently, studies demonstrated that the same co-activators were recruited by both ERs and THRs. Coactivators, such as SRC-1, transcriptional intermediary factor 2 (TIF2), and glucocorticoid receptor-interacting protein 1 (GRIP1), have the ability to interact with both the estradiol (E_2_) and TH, promoting transcription [50,51,52]. Coactivators that regulate both ERs and THRs have an identical conserved region containing helical domains of a core leucine-X-X-leucine-leucine motif, in which X represents any amino acid, and LXXLL is the structural basis for the simultaneous regulation of ERs and THRs by coactivators [50]. Furthermore, the combination of various ER with THR isoforms could influence transcription results. For instance, ligand-bound THRα1 competitively inhibits the gene transcription induced by ERα, whereas THRβ1 induces gene transcription by stimulating the activity of ERs in a CV-1 cell line derived from African green monkey kidney [53,54]. Collectively, these findings support the presumption that a competitive interaction occurs between ERs and THRs, which influences both DNA and protein levels. 

At the hormonal level, T3 and E_2_ can interact with each other, and the elevated levels of THs are able to reduce E_2_-dependent female sexual behavior [55]. In ER-positive Michigan Cancer Foundation-7 (MCF-7) breast cell line, T3 was established to regulate estrogen-induced gene expression, including PRs and transforming growth factor-alpha (TGF-α), by simulating the effect of E_2._ However, this effect is antagonized by tamoxifen, a competitive inhibitor of E_2_ [56,57]. T3 enhances the proliferative effect of E_2_ by stimulating the activity of ERE in breast cancer cells as well [58]. In this respect, Figueiredo [56] found that both E_2_ and T3 regulated almost the same genes involved in cell proliferation and protein expression after the MCF-7 cells were stimulated by E_2_ and T3. These findings suggest that the regulation of T3 may share a partly common pathway with E_2_. In addition, estrogen metabolism can be markedly altered by changes in THs [59]. Hypothyroidism may result in reduced metabolic clearance of estrone (E1) and excessive production of estriol (E3) [59,60]. Even in healthy women, cyclical fluctuations of T4 and T3 affect urinary estrogen and progesterone metabolism levels (estrone 3-glucuronide and pregnanediol 3-glucuronide) throughout the menstrual cycle [59,61]. Moreover, E_2_ can affect the level of THs and thyroid function as well. Additionally, E_2_ may regulate thyroid function by directly acting on thyroid cells [62]. The administration of double doses of estradiol valerate to the rats for six weeks induced thyroid hypoactivity, with elevated serum TSH level and decreased serum T3 and T4 levels [63,64]. High E_2_ levels in women during fresh embryo transfer would increase the risk of thyroid dysfunction in their children [62]. 

Therefore, we recommend that THs and thyroid function changes should be monitored in women suffering from gynecological diseases related to high E_2_ levels, such as endometrial cancer and endometrial hyperplasia. If necessary, THs replenishment should be performed to attenuate thyroid function damage. 

### 2.2. Crosstalk between THs and Progesterone 

PRs is another member of the steroid hormone receptor superfamily, participating in the maintenance of normal reproductive behavior and embryo development, and mediating progesterone effects [65,66]. In ewe, the distribution of THRα and PRs overlap in the diencephalon area that controls reproduction, especially in the periventricular part of the paraventricular, arcuate, and ventrolateral ventromedial nuclei [67]. Hence, PRs and THRs likely regulate the same types of cells, providing evidence that TH and progesterone co-regulate the reproductive regions of the brain. However, TRE and progesterone response elements (PREs) do not share the same half-site sequence; thus, THRs cannot recognize PRE [68]. Therefore, THRs could not interfere with PRs through PRE. However, a number of potential ERE sequences were found in the mouse PR gene sequence ranging from position −1400 to +700; thus, THRs were presumed to be able to bind to T3 via interacting with the incomplete ERE in the PRs promoter, regulating its transactivation [27]. Besides incomplete ERE initiation, the interaction between PRs and THRs also involves coactivators, similarly to the interaction between ERs and THRs. Zhang et al. [68] revealed that ligand-occupied THRα or THRβ strongly suppressed the transactivation of the progesterone-responsive reporter gene by endogenous PRs in the human-breast-cancer T47D cell line. The suppression of the progesterone-responsive reporter gene was induced via co-activator(s) that interacted with PRs or THRs [68]. Thereafter, more studies evidenced that SRC-1 and SRC-1E retained the coactivator function to enhance the ligand-dependent transactivation of PRs and THRs on the cognate response elements [69,70]. These regulatory modes may be among the mechanisms through which PRs and THRs induce gene activation and exert regulatory function on reproductive behavior. The mechanisms underlying the coordination between PRs and THRs contribute to the maintenance of the normal reproductive status. 

Clinically, progesterone and its derivatives, e.g., medroxyprogesterone and levonorgestrel, are commonly used for contraception, pregnancy maintenance, or treatment of gynecological disorders [71]. The long-term administration of progestin drugs could significantly elevate the levels of free thyroxine (FT4) and the FT4/ free triiodothyronine (FT3) ratio via suppressing the action of the deiodinases [72,73]. The application of progesterone during the first trimester of pregnancy elevates the FT4 levels and may protect the fetus from adverse consequences associated with low FT4 [74,75]. Similar to the use of complementary THs, progesterone intake could sustain appropriate maternal T4 levels and reduce the risk of neurodevelopmental defects in the newborn [74,75]. Accumulating evidence has suggested the existence of a crosstalk between THs and progesterone.

### 2.3. Crosstalk between THs and Androgens 

Testosterone can bind to ARs, and the more powerful testosterone metabolite, 5α-dihydrotestosterone (DHT), is able to activate target gene transcription [76]. Until recently, studies on the molecular mechanisms of AR actions have focused primarily on male reproductive organs [77,78,79] and female ovaries. The expression of androgen receptor-associated protein 70 (ARA70), a ligand-dependent coactivator of AR, may be involved in the progression of ovarian cancer [80]. The expression of ARA70 is upregulated by T3, thus interfering with the THR/TRE complex formation and negatively regulating T3 signaling [81]. The involvement of ARA70 in the T3-dependent signal transduction is of considerable significance in steroid crosstalk and the progress of ovarian cancer. However, whether or not abnormal expression of ARA70 occurs in patients with hyperthyroidism and hypothyroidism remains unclear [81]. In the brain and Harderian gland of amphibians, T3 promotes the expression of AR mRNA, which stimulates the male and female reproductive hormone systems [82,83,84]. Androgens, in turn, are also able to influence THs and THRs. Ovarian exposure to different concentrations of androgens can reduce THRB and iodothyronine deiodinase 3 (DIO3) expression, possibly attenuating the effect of TH on ovarian regulation [85]. In the pituitary cytoplasm of female mice, androgens inhibit the mRNA levels of the TSHα and TSHβ subunits, thereby interfering with TSH synthesis and secretion [86]. Moreover, the THs level in a mice model of Grave’s disease (GD) was significantly reduced by DHT treatment, thus confirming the protective effect of DHT against GD [87]. 

Levothyroxine treatment of polycystic ovarian syndrome (PCOS) patients with hypothyroidism decreased the total and free testosterone levels, thus alleviating their androgenic symptoms [88]. In addition, the reduction in the applied doses of levothyroxine and monitoring the serum THs levels during androgen treatment is critically important for hypothyroidism patients with breast cancer, because normal-dose therapy may induce hyperthyroidism [89], which is different from the impact of THs on estrogen. The use of estrogen by women for contraception or as hormone-replacement therapy in menopausal women may increase the bodily demand for levothyroxine [90,91]. The aforementioned findings suggest that the dysregulation of androgens and TH are closely associated with a variety of disorders [88,89]. 

### 2.4. Crosstalk between THs and Glucocorticoids

GRs belong to the steroid/thyroid hormone receptor superfamily as well, and they are activated by glucocorticoids [20,92]. Similar to the steroid hormones mentioned above, the stimulation of the hormone is converted into a transcriptional signal after the combination of glucocorticoids with GRs. GRs include receptors α and β. GRα, which is expressed in the cytoplasm, is the only receptor that binds to glucocorticoids, exerting transcriptional activity. Different from the transcriptional inhibition induced by THRs in the absence of a ligand, GRs that are not occupied by a ligand do not suppress transcriptional effects. In the absence of ligands, GRα forms a multi-protein complex with the heat-shock protein 90 (HSP90) and HSP70, as well as other factors to prevent the migration to the nucleus to suppress transcription [93]. After binding to glucocorticoids, the glucocorticoids–GRα multi-protein complex is quickly transferred to the nucleus, where it binds to GRE and regulates the transcription of target genes [94,95,96]. 

Thyroid dysfunction has been found to be associated with glucocorticoids and GR, and the majority of human research has been focused on elucidating the relationship between TH/glucocorticoid crosstalk and the function of the hippocampus. However, a limited number of studies have investigated the correlation between TH/glucocorticoid crosstalk and reproductive-system activities. An earlier study revealed the existence of a cryptic GRE upstream of TRE that had the ability to mediate a strong synergism between THRs and GRs, interfering with the transcription in a human T47D mammary carcinoma cell line [97]. In endometrial tumors with high ERs expression, GRs can recruit ERE. Due to the high similarity of TRE and ERE, GRs have the potential to bind to TRE and exhibit overlapping functions with THRs [98]. The results of these studies indicate that the crosstalk between GRs and THRs in the presence of glucocorticoids and TH is attained by synergistic or competitive regulation of DNA binding sites, resulting in specific changes in the response of each target gene. The intracellular mechanisms of the actions of GRs and THRs are the result of the interaction between glucocorticoids and THs [99]. 

Glucocorticoids not only inhibit the release of GnRH from the hypothalamus but also suppress the synthesis and release of gonadotropin and testosterone in the pituitary gland and gonads, affecting gametogenesis, sexual behavior, and fetal development [100]. In an earlier investigation, both adrenalectomy and low levels of glucocorticoids did not significantly influence the pituitary and thyroid functions in pregnant rats [101]. However, the treatment of the pregnant animals with glucocorticoids at risk of preterm delivery increased the concentration of fetal plasma T3, reversed triiodothyronine (rT3), and TSH and affected fetal thyroid axis development [101,102]. The results of these studies imply that glucocorticoids and TH crosstalk are essential to neurodevelopment. However, the mechanism underlying the interactions between glucocorticoids and TH on the female reproductive system needs further exploration.

## 3. Nongenomic Effects of THs

The other way of THs influence the female reproductive system is via nongenomic effects, in which THs stimulate the mitochondria and cytoskeleton in the cytoplasm or integrin receptors on the cell membrane to target extra-nuclear signaling regulated by hormone-mediated rapid cellular response without involving the intra-nuclear THRs induced genomic effects by T3 [19,103,104]. To date, few reports have been published on the mitochondria and cytoskeleton in cells of the reproductive system, whereas the majority of the nongenomic studies has been focused on the association between THs and integrin receptors. 

Integrin αvβ3 possessing S1- and S2-binding domains has no structural homology with THRs. It is different from the membrane G-protein-coupled receptors (GPCRs) peptide or protein reproductive hormones (Figure 2). In human glioblastoma U-87 MG cells, the S1 domain of integrin αvβ3 specifically recognizes T3 and activates the phosphatidylinositol-3-hydroxykinase (PI3K) signaling pathway through Src kinase, mediating downstream cytoplasmic THR shuttling to the nucleus and inducing the expression of the hypoxia-inducible factor 1-alpha (HIF-1α) (Figure 2). The S2 domain is involved in the regulation of the extracellular-signal-regulated kinases 1/2 (ERK 1/2) pathway. The activation of the ERK1/2 signaling triggers the downstream mitogen-activated protein kinase (MAPK). In turn, this reaction promotes the phosphorylation of the signal transducer and activator of transcription-1α (STAT-1α), STAT3, p53, and ERs, leading to tumor cell proliferation and the nuclear uptake of THRβ1 from the cytoplasm [19,105,106]. Although integrin and peptide and protein hormone receptors exert effects via different pathways, they can generate crosstalk by activating the same signaling pathway [107]. The signaling pathways activated by THs and some reproductive hormones are displayed in Figure 2.

Peptide and protein reproductive hormones are different from steroid hormones. They bind to the cell surface membrane receptors, activate signal transduction pathways through nongenomic effects, and continuously transmit information quickly and briefly [46]. The concrete mechanism underlying the mutual interference among TH and peptide and protein hormones through the same signal transduction pathway is not thoroughly understood.

### 3.1. Crosstalk between THs and FSH and LH

TSH, follicle-stimulating hormone (FSH), and luteinizing hormone (LH) are glycoprotein hormones that are synthesized in the anterior pituitary gland. They participate in reproductive development and the presentation of secondary sexual characteristics [108]. All three peptide hormones consist of a common alpha subunit and specific beta subunits [109]. They exert effects by binding to their respective GPCRs [109]. By using highly purified bovine LH and TSH and their subunits, Williams [110] found that a high concentration of LH affinity-purified preparations retained the TSHRs binding activity in rat testis. Accordingly, the investigator presumed that LH preparations and other hormones containing α-subunits had inherent TSHR binding activities, albeit with significantly lower activity. This finding implies that LH and FSH combine with TSHRs and induce the initiation of the next step of TSHR signaling. Another experiment revealed that the expression levels of the α subunit mRNA of FSH, LH, and TSH in the pituitary gland were increased in rats suffering from hypothyroidism, but the alterations were reversed after acute or chronic T3 treatment [111]. This result indicates that T3 directly regulates the α subunit of FSH and LH. Both T3 and TSH have the capability of regulating and binding to hormones containing α-subunit, including FSH and LH. 

FSH and LH synergistically stimulate FSHRs and LHRs in the ovary to secrete estrogen, which is important for follicle development and maturation. T3 stimulation alone cannot promote the development of follicles, but it promotes the growth of the follicles after binding to T3 and FSH in rats [112,113]. Additionally, T3 enhances the follicle-stimulating effect of FSH by upregulating FSHR mRNA expression, which is dependent on the growth differentiation factor-9 (GDF-9) pathway [112]. In ovarian granulosa cells, T3 and FSH mediated and downregulated the expression of factor-associated suicide FAS (Fas/FasL) and increased the level of antiapoptotic protein X-linked inhibitor of apoptosis protein (XIAP) by activating Src and modulating the PI3K/protein kinase B (Akt) signaling pathway, ultimately inhibiting the apoptosis of the granular cells [114]. In addition, positive TREs were reported to be present in regions upstream of the transcription start site of the LHRs gene in mouse Leydig tumor cells-1 (mLTC-1). T3 treatment increased the expression of LHRs gene and promoted ligand binding [115]. These findings provide novel evidence that the interaction between THs, FSH, and LH likely is involved in the development and regulation of germ cells.

Hyperthyroidism is known to increase the secretion of LH, but the influence on FSH is still controversial [116]. This condition is considered to affect fertility by altering FSH and LH, since both of them are involved in ovulation induction [117,118]. Taken together, the structural similarities of TSH, FSH, and LH increase the probability of their interactions. In addition, THs have the capabilities of regulating the development of germ cells and even interfering with the ovulation process by affecting FSH and LH. Hence, the influence of thyroid dysfunction on FSH and LH, as well as its intervention in the process of mammalian reproduction, is understandable [119].

### 3.2. Crosstalk between THs and GnRH

Gonadotrophin-releasing hormone (GnRH) is a peptide hormone released by the hypothalamus that stimulates the production and secretion of LH and FSH in the anterior pituitary gland [120]. In addition to its regulation of gonadotropin effects, GnRH also interacts with THs. TSH release in adult amphibians is stimulated by GnRH [121]. In the human body, GnRH regulates the thyroid gland and THs inconsistently. Whether GnRH treatment induces thyroid dysfunction remains controversial. A study showed that more than 70% of the patients with the symptoms of central precocious puberty developed thyroid function impairment after using GnRH agonist for 12.37 months [122]. However, the results obtained by Massart et al. [123] revealed that GnRH agonists did not induce thyroid dysfunction in euthyroid subjects, although the GnRH agonist slightly suppressed the thyroid secretion. The differences in the aforementioned studies may be ascribed to the discrepancies in the GnRH agonist treatment duration and the TH-level monitoring time [123]. To minimize uncertainty, the investigator suggested that the serum levels of TSH, FT4, FT3, and thyroid antibodies should be evaluated after one year of GnRH therapy.

A large number of animal experiments have been conducted to establish the association between TH and GnRH. THRs protein and mRNA have been discovered in GnRH neurons in many species [124,125]. Notably, GnRH has been confirmed to mediate the effect of TH on the reproductive axis of vertebrates. High-frequency fluctuations of GnRH were distinctly observed in ewes with thyroidectomy; the sexual activity of the ewes was invariably maintained, and the estrus period could not be ended [126]. T4 injection into the cerebrospinal fluid or T4 implantation in the ventromedial preoptic area of the middle hypothalamus ended the estrus, but the sexual maturity was impeded via suppression of the expression of the terminal nerve GnRH [127]. This evidence indicates that TH is necessary for the regulatory function of GnRH exerted on the estrous cycle in animals. Although TH is known to interfere with the reproductive process by affecting the level of GnRH, the concrete molecular mechanism is still unclear; thus, it warrants further research.

### 3.3. Crosstalk between THs and Prolactin

Prolactin is a protein hormone secreted by the anterior pituitary gland [128]. It promotes mammary gland development and initiates lactation [129]. In hyperthyroidism, the prolactin secretion from the pituitary is inhibited, while the serum prolactin level is increased [130,131]. Approximately 40% of the patients with hypothyroidism simultaneously suffer from hyperprolactinemia. Many hypotheses have been proposed to explain the reasons for the elevated levels of prolactin in hypothyroid patients. In patients with primary hypothyroidism, the elimination rate of prolactin is usually lower, which augments the concentrations of prolactin in the blood [132,133]. In an in vitro study, the human prolactin gene was found to have two significant T3-responsive motif in the primary regulatory region (the proximal promoter), which separately mediated potent inhibition and weak promotion of T3 [134]. Altogether, the combination of these two reaction domains exerts an inhibitory effect; that is, T3 negatively regulates the human prolactin promoter [134]. After treatment with T3 (10 nM), THRs became bound to the activator protein-1 (AP-1) response element of the prolactin promoter (61/54) and inhibited the transcriptional activation of AP-1 in human prolactin [135]. Moreover, T3 was shown to inhibit prolactin mRNA levels and prolactin synthesis in a dose-dependent manner [136,137]. In addition, TRH secreted from the hypothalamus was delivered to the pituitary gland, where it induced prolactin synthesis and secretion [138,139]. In this process, TRH stimulates phospholipase C (PLC) production by activating the Gq-protein-coupled TRH receptor, followed by activation of protein kinase C (PKC)-dependent ERK signaling pathway, which promotes the expression of prolactin genes [140]. TRH also activates ERK by phosphorylating generic sarcoma homology 2 domain containing (Shc) protein to promote prolactin gene transcription [140]. As a result, TRH promotes the synthesis of prolactin and increases serum prolactin level as a stimulating factor of the pituitary gland. Collectively, both an increased TRH level and decreased T3 level in the hypothyroidism are associated with elevated expression of the prolactin gene, promoting prolactin synthesis. Therefore, the treatment of this kind of hyperprolactinemia requires administering TH drugs. 

In addition, the treatment of hyperprolactinemia induced by other causes, such as pituitary tumors or bromocriptine (a postsynaptic dopamine-D2-like receptors agonist) administration, can inhibit the levels of TSH and decrease the size of the goiter, while sustaining clinical euthyroidism [141]. Clinical evidence has shown that hyperprolactinemia therapy with thyroid hormone drugs or bromocriptine facilitates the recovery of patients [142]. The effectiveness of this treatment reveals the advantages of therapeutically applying hormonal crosstalk.

### 3.4. Crosstalk between THs and Oxytocin

Oxytocin is not only a bioactive protein that stimulates prolactin release and lactation; it also interferes with the synthesis and release of TRH and suppresses TRH-induced TSH release, decreasing the plasma levels of TSH and TH through its effects on the central nervous system [143,144]. The promotor gene of oxytocin and oxytocin receptor contain composite hormone response elements, which respond to both estrogen and THs and are able to be elicited by ERs and THRs [145]. THs stimulated the expression of the oxytocin gene by directly affecting the activity of its promoter through the TH–THRs complex [146]. On the other hand, THs also directly promoted the transcription of oxytocin mRNA by interacting with the TRE of the oxytocin gene [146]. This process constitutes a stable TH–oxytocin feedback regulation. In vivo, high-dose administration of T3 (500 μg/kg/day) increased the expression of oxytocin mRNA in the paraventricular nucleus of the hypothalamus (PVN) in ovariectomized rats, whereas a high dose of T3 combined with E_2_ reduced the oxytocin mRNA level [147]. It suggests that a competitive interaction exists between liganded THRs and ERs on the oxytocin gene, which may be a vital mechanism by which TH and E2 regulate the oxytocin gene expression in PVN.

In addition to the oxytocin gene, the oxytocin receptor gene is also regulated by THs and E2. E2 stimulates both oxytocin and progesterone receptors in the ventromedial and preoptic regions of the hypothalamus, whereas THs inhibit the E2-mediated behaviors. Ligand-occupied THRs interfere with the ERα-mediated transcription induction in the oxytocin receptor gene promoter, and this impact varies with different cell lines and subtypes of THRs, thus causing the diversity in gene regulation to achieve endocrine integration [54]. Moreover, oxytocin and its receptor are also involved in regulating and activating downstream signaling cascades in a nongenomic manner, as well, such as MAPK, PKC, or PLC [148], and the pathways play a vital role in the regulation of TH and integrin receptor activation (Figure 2). For example, oxytocin is involved in regulating prostaglandins release in ovarian cells and modulating endometrium through PLC- and ERK-related MAPK signaling pathways [149,150,151,152].

Overall, both the oxytocin gene and its receptors are part of the targets of TH, and TH and THRs act as physiological mediators and regulators involved in the modulation of oxytocin in reproductive behavior via both genomic and nongenomic mechanisms.

## 4. Clinical Consequences of the Crosstalk between THs and Reproductive Hormones 

Studies have shown that THs affect the secretion and the action of reproductive hormones via multiple pathways. Specifically, THs act directly on the ovary, uterus, and placenta through their interactions with estrogen, progesterone, androgens, FSH, LH, and prolactin. They also influence the release of GnRH in the hypothalamic–pituitary–gonadal (HPG) axis [91,153,154]. The basic modulation processes of these hormones via the HPG axis are highly similar among mammals, but the consequences of regulation vary across different species and different thyroid states in terms of the circumstances of hypothyroidism and hyperthyroidism. 

In humans, the changes of the levels of FSH and LH are usually inconsistent in women with hypothyroid or hyperthyroid. Hyperthyroidism is associated with increased levels of total E2, testosterone, and progesterone, effects that are contrary to the abnormalities caused by hypothyroidism [155,156,157]. However, the fluctuations in serum reproductive hormones in hypothyroid and hyperthyroid rat models were totally different [158,159]. Therefore, the regulation of THs on reproductive hormones in mammals is complex. Thereby, it is critical to explore the effects of THs on reproductive hormones, which is conducive to apprehending the etiology of reproductive disorders. The regulatory effects of hyperthyroidism and hypothyroidism on female reproductive hormones are illustrated in Figure 3.

The occurrence of female reproductive diseases is considered to be related to reproductive hormone disorders, but recent studies have shown that the regulatory interference of THs in the synthesis and activities of reproductive hormones may also be a cause of reproductive diseases. In both the uterus and ovary, TH seems to act as a mitotic and pro-survival factor, promoting the proliferation of endometrial decidua cells and the growth and survival of follicles and granular luteum [160,161,162,163]. THs and their receptors play an important role in the maintenance of the normal structure and function of the uterus and ovaries [38,164]. As a result, thyroid dysfunction is associated with gynecological diseases, reproductive tumors, and even infertility [165,166,167]. 

For instance, women with menstrual disorders and sub-fertility are more susceptible to hypothyroidism [168], and more cases of chronic pelvic pain have been established in endometriosis patients with thyroid dysfunction than in patients without thyroid dysfunction [162]. In addition, PCOS patients have a higher risk of autoimmune thyroid disease (AITD) compared with normal women [169,170,171]. Furthermore, hypothyroidism seems to be associated with poor prognosis in patients with endometrial cancer [172]. Some researchers even proposed that TSH could be used as a new prognostic parameter for the survival of patients with endometrial cancer [173]. Based on this evidence, the alteration of THs and their receptors could be considered a diagnostic indicator and a treatment target for reproductive diseases. 

## 5. Conclusions 

In the review, we have summarized and analyzed, for the first time, the existing research findings on the crosstalk between THs and reproduction-associated hormones and their subsequent regulation. The modulation of THs involves an intricate network, including a self-regulating loop and interactions with various peripheral steroids and peptide hormones. In the past decades, accumulated evidence has shown that THs and reproductive hormones, including E2, progesterone, glucocorticoids, FSH, LH, prolactin, and oxytocin, jointly regulate the reproductive system processes, thus predetermining their involvement in the occurrence of reproductive diseases. Through binding to intracellular THRs or hormonal receptors or membrane receptors, THs regulate the behavioral and physiological effects of the reproductive hormones by activating the gene networks.

As a result of this complex crosstalk and interactions, THs interfere with the synthesis, secretion, and effects of other hormones and can even affect the whole hormonal balance. Therefore, they are among the pivotal factors involved not only in the regulation of female reproductive processes but also in the occurrence of hormone-associated diseases. More studies are warranted on this topic, since the underlying molecular mechanisms are not yet thoroughly understood. Elucidating the interactive mechanism among THs and THRs and reproductive hormones will contribute to apprehend the etiology of female reproductive diseases and provide novel insights and possibilities for the treatment of the gynecological disorders.

## Figures and Tables

**Figure 1 ijms-23-02708-f001:**
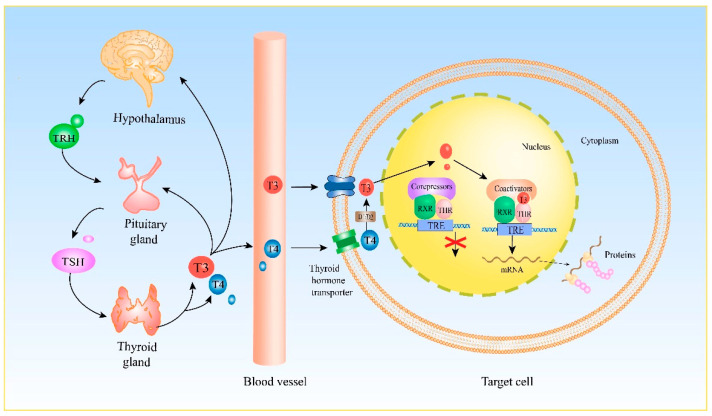
Secretion of TH and its genomic effect on target cells. T3 and T4 secreted by the thyroid gland are released into the blood and transported to the pituitary and hypothalamus. T3 and T4 enter the target cells through the TH transporter. After TH enters the nucleus, T4 is converted to T3 by D1/D2. Before the entry of T3 into the nucleus, THR heterodimerizes with RXR and binds to TRE on the DNA, and then recruits corepressors, inhibiting the transcription. After T3 enters the nucleus, it binds to the ligand-binding domain of THRs, disrupting the corepressor binding while promoting coactivator binding. Then it initiates transcription. Abbreviations: TRH, thyrotropin-releasing hormone; TSH, thyroid-stimulating hormone; T3, triiodothyronine; T4, thyroxine; D1, 1 5′-deiodinase; D2, 2 5′-deiodinase; RXR, retinoic acid receptor; THR, thyroid hormone receptor; TRE, thyroid-hormone response element. ↓, Stimulatory function.

**Figure 2 ijms-23-02708-f002:**
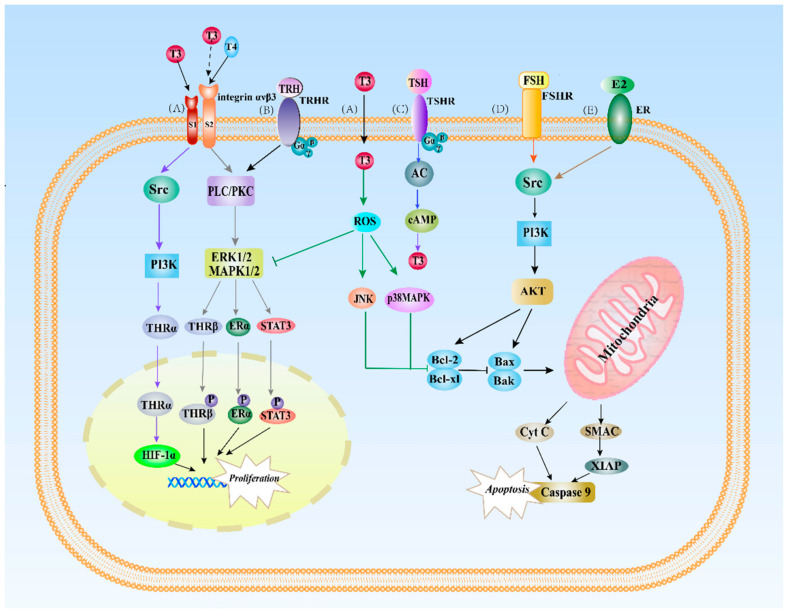
Nongenomic actions of TH and reproductive hormones in the female reproductive system. (A) The nongenomic effects of THs are initiated by integrin α V β3 in the plasma membrane. T3 interacts with the S1 domain of integrin α V β3 and activates the PI3K signaling pathway via Src kinase, resulting in the transport of THRα from the cytoplasm to the nucleus and increasing HIF-1 α expression. T4 and T3 also interact with the S2 domain of integrin α V β3 to activate ERK1/2 signaling, leading to phosphorylation and nuclear localization of THR β, ERα, and STAT3 to promote cell proliferation. T3 promotes a local proinflammatory environment and stimulates ROS generation to activate JNK and p38MAPK, enhancing ectopic cell proliferation and angiogenesis; (B) TRH stimulates the phospholipase C (PLC)/protein kinase C (PKC)-dependent pathway to activate ERK signaling and promotes prolactin gene expression by activating Gq protein coupled with the TRH receptor. (C) TSH binds to TSHR, promoting the synthesis and release of TH through the cAMP cascade effect; (D) FSH binds to FSHR and increases the levels of the antiapoptotic protein XIAP and caspase-9 by activating Src and via the PI3K/Akt signaling pathway, ultimately inhibiting apoptosis; (E) ER also activates Src and PI3K/Akt signaling pathways to promote cell proliferation and anti-apoptosis in the female reproductive system. Abbreviations: Src, proto-oncogene tyrosine-protein kinase; PI3K, phosphatidylinositol-3-kinase; PLC, phosphorylase C; PKC, protein kinase c; ERK, extracellular-signal-regulated kinases 1/2; MAPK, mitogen-activated protein kinase; STAT3, signal transducer and activator of transcription 3; ROS, reactive oxygen species; JNK, c-Jun N-terminal kinase; AKT (protein kinase B); cAMP, cyclic adenosine monophosphate; Bcl-2, B-cell lymphoma-2 Bax, Bcl2-associated X protein; Bak, Bcl2 antagonist/killer 1; Cyt C, cytochrome C; SMAC, second mitochondria-derived activator of caspase; XIAP, X-linked inhibitor of apoptosis protein. ↓, Stimulatory modification; ⊥, Inibitory modification. Different colors indicate signaling pathways validated in different literatures.

**Figure 3 ijms-23-02708-f003:**
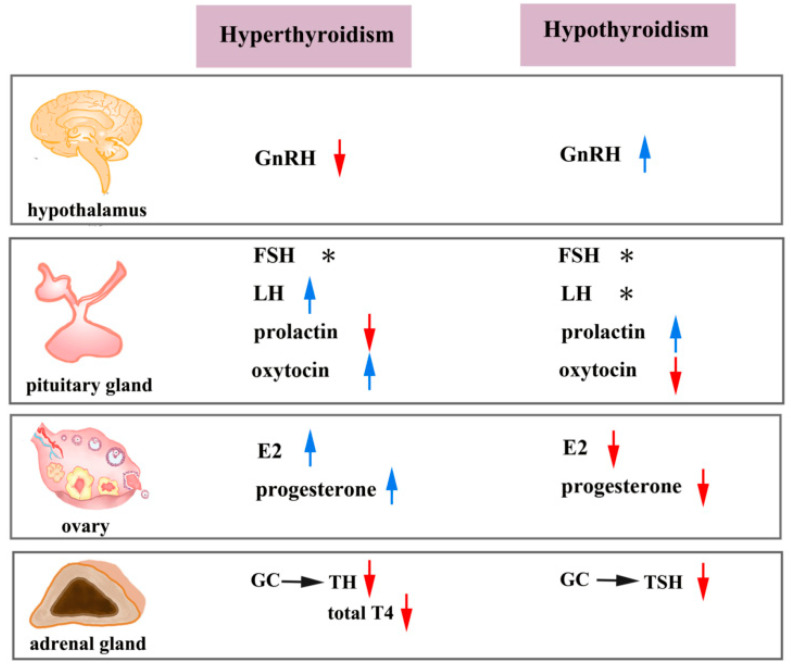
Effects of hyperthyroidism and hypothyroidism on serum hormone levels in women. Thyroid dysfunction can result in changes in the reproductive hormone secretion. The horizontal arrow represents upregulation, whereas the downward arrow indicates that hyperthyroidism and hypothyroidism decrease serum hormone levels. The upward arrow indicates that hyperthyroidism and hypothyroidism increase serum hormone levels. The asterisk indicates that the changes of hormone levels are inconsistent among various sources in the literature. GnRH, gonadotropin-releasing hormone; FSH, follicle-stimulating hormone; LH, luteinizing hormone; E2, estradiol; GC, glucocorticoid; TH, thyroid hormone; T4, thyroxine; TSH, thyroid-stimulating hormone. The red arrow indicates decreased hormone levels. The blue arrows indicate increased hormone levels. Asterisks indicate inconsistencies in the literature.

**Table 1 ijms-23-02708-t001:** Nuclear receptors sharing the same DNA consensus sequence and their role in female reproduction. HRE usually consists of two consistent palindromes (mirror images) or directly repeated hexamers. The palindromes are also referred to as “half-site”, because each site binds to a monomer of the receptor. Typically, TRE, PPARE, RXRE, VDRE, and ERE in the promoter regions of downstream genes contain two half-site sequences AGGTCA in a palindromic repeat, and the direct or inverted repeat arrangement can be recognized by their receptor. GRE, PRE, MRE, and ARE share the same two half-site sequences AGAACA in a palindromic repeat. TRE, thyroid hormone response elements; PPARE, peroxisome proliferator-activated response element, RXRE, retinoid X response element; VDE, vitamin D response element; ARE, androgen response element; GRE, glucocorticoid response element; MRE, mineralocorticoid response element; PRE, progesterone response element. R = adenine (A)/guanine (G), K = G/thymine (T), W = A/T, V = A/cytosine (C)/G, N = A/C/G/T.

Receptor	NR Family	The Function of Nuclear Receptors in Female Reproduction	Ligand	Half-Site Sequence
THRA	NR1A1	THRA contributes to ovarian follicular development [31].	Thyroid hormone	RGGTVA
THRB	NR1A2	Loss of THRβ expression is related to the rise in the 5-year survival rate of endometrial cancer [32].		
RARA	NR1B1	RARA might regulate vascular formation in the human endometrium [33].	Retinoic acid	AGTTCA
RARB	NR1B2	Promoter methylation of RARB and BRCA1 predicts a worse prognosis in cervical cancer [34].		
RARG	NR1B3	Fertile: RXRB/RXRG double knockout are also fertile [35].		
PPARA	NR1C1	PPARα activation influences endometrial cell growth and VEGF secretion [36].	Eicosapentaenoic acid	AGGTCA
PPARD	NR1C2	PPARD expression at the implantation sites requires the presence of an active blastocyst and may play an essential role for blastocyst implantation [37].		
PPARG	NR1C3	PPARγ activation reduces proliferation of endometrial cells via regulation of PTEN [36].		
VDR	NR1I1	VDR pathways are modulated in normal and disease endometrium by activation of vitamin-D-regulated genes [38].	Vitamin D3	RGKTCA
ER1	NR3A1	Early female endometriosis is mediated by immune-mediated crosstalk between ER1 and IL-6 [39]. Female mice lacking ER1 are infertile due to impaired ovarian and uterine functions [40].	Estrogen	RGGTCA
ER2	NR3A2	Female mice lacking ERβ are sub-fertile due to ovarian defects [40].		
GR	NR3C1	GR is required to establish the necessary cellular context for maintaining normal uterine biology and fertility through the regulation of uterine-specific actions [41].	Cortisone	AGAACA
MR	NR3C2	MR may play an endocrine/paracrine/autocrine role in the bovine ovary [42].	Aldosterone	AGAACA
PR	NR3C3	PR binds the genomic regions of genes regulating critical processes in uterine receptivity and function [43]	Progesterone	AGAACA
AR	NR3C4	AR and AR signaling have a decisive role in the differentiation of human endometrial stromal cells into decidual cells [44].	Testosterone	AGAACA

## Data Availability

Not applicable.

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
