# Peer review of "A New Perspective on Thyroid Hormones: Crosstalk with Reproductive Hormones in Females"

_ijms, 2022, doi:10.3390/ijms23052708_

Round 1
Reviewer 1 Report
The manuscript ijms-1526657 concerns the interaction between thyroid and reproduction hormones in females. Mechanisms of crosstalk between endocrine systems and potential functional implications are important topics especially in the context of endocrine disrupting compounds. The topic is suitable to the journal. I have listed some suggestions below:
- Line 147 – the text is not visible because of Figure 2.
- As estrogen and progesterone are described considering their interaction with TH, androgens couls also be mentioned.
- Are there any literature data considering the interaction between thyroid hormones and other estrogens (estrone and estriol)?
- A small chapter on interactions in the context of endocrine disrupting compounds is worth considering in the manuscript.
- Minor English editing is needed.
Reviewer 2 Report
This is a paper where authors summarized the interactive patterns of between THs and reproductive hormones and their receptors to provide a new perspective for clinical diag4 nosis and treatment of reproductive diseases.
Although this non structured review contains very useful information on the subject, it is difficult to follow.
I would recommend if authors’ target is publication, to rewrite it in a more structured way, engaging all the relevant sections, including clear rationale and aim, gap in the literature, search strategy, search results, main and secondary endpoints and findings, final conclusion, limitations and interpretation of their findings.
